# Investigation of Stabilized Amorphous Solid Dispersions to Improve Oral Olaparib Absorption

**DOI:** 10.3390/pharmaceutics16070958

**Published:** 2024-07-19

**Authors:** Taehan Yun, Sumin Lee, Seowan Yun, Daeyeong Cho, Kyuho Bang, Kyeongsoo Kim

**Affiliations:** Department of Pharmaceutical Engineering, Gyeongsang National University, 33 Dongjin-ro, Jinju 52725, Republic of Korea; xogks7702@naver.com (T.Y.); m8121@naver.com (S.L.); wyjm2013@naver.com (S.Y.); daey2059@naver.com (D.C.)

**Keywords:** olaparib, solid dispersion, HPMC, solubility, bioavailability

## Abstract

In this study, we investigated the formulation of stable solid dispersions to enhance the bioavailability of olaparib (OLA), a therapeutic agent for ovarian cancer and breast cancer characterized as a BCS class IV drug with low solubility and low permeability. Various polymers were screened based on solubility tests, and OLA-loaded solid dispersions were prepared using spray drying. The physicochemical properties of these dispersions were investigated via scanning electron microscopy (SEM), differential scanning calorimetry (DSC), powder X-ray diffraction (PXRD), and Fourier Transform Infrared Spectroscopy (FT-IR). Subsequent dissolution tests, along with assessments of morphological and crystallinity changes in aqueous solutions, led to the selection of a hypromellose (HPMC)-based OLA solid dispersion as the optimal formulation. HPMC was effective at maintaining the supersaturation of OLA in aqueous solutions and exhibited a stable amorphous state without recrystallization. In an in vivo study, this HPMC-based OLA solid dispersion significantly enhanced bioavailability, increasing AUC_0–24_ by 4.19-fold and C_max_ by more than 10.68-fold compared to OLA drug powder (crystalline OLA). Our results highlight the effectiveness of HPMC-based solid dispersions in enhancing the oral bioavailability of OLA and suggest that they could be an effective tool for the development of oral drug formulations.

## 1. Introduction

OLA is a potent inhibitor of poly ADP ribose polymerase (PARP), a drug approved as a first-in-class for PARPi [1], and is widely used to treat cancers associated with BRCA1 and BRCA2 mutations, such as ovarian cancer, breast cancer, and prostate cancer [2]. The commercial product of OLA is Lynparza tablets, which are administered orally as two 150 mg tablets (300 mg) twice a day, with a total daily dose of 600 mg [3]. Despite its many therapeutic benefits, OLA is classified as a class IV molecule according to the Biopharmaceutics Classification System (BCS) due to its low solubility and low permeability [4]. OLA has a low solubility of approximately 0.1 mg/mL in aqueous solution, with a basic pKa of −1.25 and an acidic pKa of 12.07 [5].

Generally, poorly water-soluble drugs, such as OLA, have low oral absorption and bioavailability due to their low solubility, leading to increased administered doses and decreased patient compliance [6]. Therefore, improving the solubility of poorly soluble drugs is important in the development of oral dosage forms [7]. To enhance the solubility of poorly soluble drugs such as OLA, various techniques have been devised, including particle size reduction, prodrugs, polymeric nanoparticles, lipoidal microspheres, inclusion complexes, salt formation, and lipid-based formulations [8,9]. However, these methods have several limitations, including low drug loading capacity, complex physical structures, instability, potential toxicity of materials, and changes in drug distribution and elimination [9].

Solid dispersion systems have been widely used to enhance the solubility and bioavailability of water-insoluble drugs [10]. Additionally, extensive research has been conducted on solid dispersions for BCS class IV drugs like OLA [11,12,13,14]. Solid dispersion is defined as the dispersion of one or more drugs at the molecular level within a polymer carrier matrix and can be prepared using methods such as the melting method, the solvent-evaporation method, and the solvent-wetting method [15]. Solid dispersions have the advantage of not requiring special equipment compared to other solubilization technologies, and the preparation process is simpler than that of techniques using materials such as nanoparticles or nano-emulsions [16]. Additionally, many poorly soluble drugs have been approved and marketed through solid dispersions [17]. One of the fundamental principles of solid dispersion formulation is to achieve an amorphous state, which is considered to be more soluble than the crystalline state [18]. Therefore, the selection of polymers that effectively enhance solubility and provide stability to the amorphous form is important in solid dispersion systems [19].

To date, solubility studies of OLA have only reported on self-microemulsifying drug delivery systems (SMEDDSs) and nano-based formulations, with studies on OLA-loaded solid dispersions (OLA-SDs) being difficult to find [20,21]. In this study, we attempted to develop OLA-SDs by selecting appropriate polymers that can enhance solubility and prevent recrystallization. After determining the solubility characteristics of OLA in aqueous solution, the solubility of OLA in various polymers was investigated. Subsequently, OLA-SDs were prepared using the selected polymers through spray drying. The physicochemical properties of the prepared OLA-SDs were assessed using SEM, DSC, PXRD, and FT-IR. The dissolution rate and surface characteristics of the OLA-solid dispersion in aqueous solution were evaluated. Finally, the improvement in bioavailability was evaluated by comparing crystalline OLA, amorphous OLA without polymers, and OLA-SD through in vivo pharmacokinetics (PK) studies in rats. A concise overview of the research design is shown in Figure 1.

## 2. Materials and Methods

### 2.1. Materials

OLA was purchased from Olon S.p.A. (Rodano, Italy). Hydroxypropyl-β-cyclodextrin (HP-β-CD), α-cyclodextrin (α-CD), and γ-cyclodextrin (γ-CD) were supplied by Ashland Inc. (Wilmington, DE, USA). Poly (ethyl acrylate, methyl methacrylate, trimethylammonioethyl methacrylate chloride) with a ratio of 1:2:0.2 (abbreviated as Eudragit RL100), 1:2:0.1 (abbreviated as Eudragit RS100), and Poly(butyl methacrylate, (2-dimethylaminoethyl) methacrylate, methyl methacrylate) with a ratio of 1:2:1 (abbreviated as Eudragit E PO) were obtained from Evonik (Essen, Germany). Carbomer homopolymer type A (Synthalen LP) and Carbomer homopolymer type B (Synthalen E83P) were sourced from 3V Sigma (Georgetown, SC, USA). Povidone (PVP), Copovidone (Kollidon VA64), and Polyvinyl Alcohol–Polyethylene Glycol Graft Copolymer (Kollicoat IR) were acquired from BASF (Ludwigshafen, Germany). Polyethylene Glycol (PEG), Maltodextrin, Dextran, Hypromellose (HPMC), Hydroxypropyl Cellulose (HPC), and Sodium Alginate were kindly provided by Hanmi Pharmaceutical Co., Ltd. (Hwaseong, Republic of Korea). Gelatin was purchased from Sigma Aldrich (Saint Louis, MO, USA). Pectin and ethanol were purchased from Daejung Chemicals (Siheung, Republic of Korea). Colloidal silica was supplied by Boryung Pharmaceutical Co., Ltd. (Seoul, Republic of Korea). The deionized water used in the laboratory was produced using a distillation device. All other chemicals were of an analytical grade.

### 2.2. HPLC Analysis Condition

An OLA concentration test was performed using HPLC (Agilent 1260 series; Agilent Technologies, Santa Clara, CA, USA) equipped with a UV-Vis detector and a high-pressure gradient pump. The analytical column was a C18 column (SunFire, 4.6 × 150 mm, 5 μm; Waters, Milford, MA, USA). The analytical column was a C18 column (SunFire, 4.6 × 150 mm, 5 μm; Waters, Milford, MA, USA). The mobile phase was composed of a mixture of 2 g/L KH_2_PO_4_ buffer and acetonitrile (65:35, *v*/*v*), with the column temperature maintained at 30 °C. The injection volume and flow rate were 10 μL and 1.0 mL/min, respectively, and the UV absorbance was set at 220 nm. The HPLC conditions for measuring the concentration of OLA in plasma after oral administration of OLA to rats were slightly adjusted from the conditions mentioned above. The HPLC analysis employed a C18 column (ZORBAX Eclipse XDB-C18, 4.6 × 250 mm, 5 μm; Agilent Technologies, Santa Clara, CA, USA), and the mobile phase consisted of 2 g/L KH_2_PO_4_ buffer (pH 4.5) and acetonitrile (50:50, *v*/*v*). The column temperature was maintained at 30 °C, with an injection volume of 10 μL and a flow rate of 1.0 mL/min. The UV absorbance was set at 213 nm [22]. Data collection and processing were performed using OpenLab CDS CS C.01.08 ChemStation software.

### 2.3. Drug Solubility Test

The solubility of OLA was evaluated in distilled water (D.W.), pH 1.2, pH 4.0, and pH 6.8 solutions. The pH 1.2 solution was prepared using 0.1 M hydrochloric acid and sodium chloride, while the pH 4.0 solution utilized a 0.05 M sodium acetate buffer solution, and the pH 6.8 solution was made by combining a 0.2 M potassium dihydrogen phosphate solution with a 0.2 M sodium hydroxide solution. To determine the saturation solubility, an excess amount of both crystalline OLA and amorphous OLA, totaling 10 mg each, was added to 1 mL of each solution. The mixtures were vortexed for 10 min and then shaken at 100 rpm in a 37 °C water bath for 72 h. After centrifugation at 13,500 rpm for 10 min, the supernatant was filtered through a 0.45 μm syringe filter to remove insoluble OLA. The resultant solution was diluted tenfold with the HPLC mobile phase, and the concentration of OLA in each solution was measured using the HPLC system mentioned in Section 2.2. Subsequently, to assess the kinetic solubility of OLA, a dissolution test was conducted using 150 mg of both crystalline and amorphous OLA in 900 mL of each aforementioned solution. This test utilized a USP dissolution apparatus II (RCZ-6N; Pharmao Industries Co., Liaoning, China), with the solution temperature adjusted to 37 ± 0.5 °C and the paddle speed set at 50 rpm. Samples (3 mL) were taken at predetermined times (0.5, 1, 2, 3, 4, 6, 12 h), filtered through a 0.45 μm syringe filter, diluted tenfold with the HPLC mobile phase, and analyzed using the same HPLC system. The preparation method for amorphous OLA is described in Section 2.5. Briefly, crystalline OLA was completely dissolved in a mixture of ethanol and D.W., and this solution was spray-dried to prepare amorphous OLA.

### 2.4. Screening of Polymers

The polymers used to select the appropriate polymer for OLA-SD were chosen from those widely utilized in solid dispersions [23,24,25,26,27,28,29,30,31,32,33,34,35,36,37]. To select the polymers for the solid dispersion system, the solubility of OLA in 1% (*w*/*v*) polymer solutions was assessed. Initially, 10 mg of crystalline OLA was added to 1 mL of each polymer solution, vortexed for 10 min, and then shaken at 100 rpm in a 37 °C water bath for 72 h. This process followed the saturation solubility evaluation method described in Section 2.3. Subsequently, to evaluate the kinetic solubility of OLA, both crystalline and amorphous forms, totaling 10 mg, were added to 10 mL of the selected polymer solutions, based on the results of the OLA saturation solubility tests. After vortexing for 10 min, the samples were shaken under the same conditions. During this 72 h period, samples were collected at predetermined intervals of 1, 6, 12, 24, 48, and 72 h, and the concentration of OLA was measured through HPLC analysis.

### 2.5. Preparation of Solid Dispersion

There are various methods for the preparation of solid dispersions, particularly the spray drying process, which is rapid, continuous, and capable of ensuring reproducibility during scale-up, making it widely used in the industrial preparation of solid dispersions [38]. Additionally, spray drying has the advantage of a faster solvent removal rate compared to other solvent-evaporation processes, which can reduce the tendency of the drug to crystallize [39]. The OLA-SDs were prepared using a spray dryer (Yamato ADL311SA; Yamato Scientific Co., Ltd., Tokyo, Japan). Based on the results of the polymer screening, 300 mg of the selected polymer was dissolved in a mixture of ethanol and distilled water (D.W.), and 150 mg of OLA was dissolved in this solution. Based on preliminary study, the ratio of OLA to polymer was selected as 1:2 (*w*/*w*), which is the ratio that can enhance the dissolution rate of OLA (Appendix A). Then, 75 mg of colloidal silica, which can enhance the yield of the solid dispersion, was uniformly suspended in the solution and subsequently spray-dried. The spray drying conditions were an inlet temperature of 85 °C, an outlet temperature of 55 °C, a feeding flow rate of 1.5 mL/min, and an atomizing air pressure of 0.1 MPa. The obtained solid dispersion was further dried by storing it in a 60 °C dry oven for 30 min, after which the LOD (Loss on Drying) was confirmed to be below 1.0%. The samples were then stored in an ethylene bag with silica gel. The detailed compositions of the spray drying solution for the preparation of OLA-SDs are shown in Table 1. The polymer structures used in the preparation of OLA-SDs are illustrated in Figure 1.

### 2.6. Physicochemical Characterization of OLA-SD

#### 2.6.1. Surface and Morphological Features

Scanning electron microscopy (Tescan-MIRA3; Tescan Korea, Seoul, Republic of Korea) was used to examine the surface and morphological features of the OLA-SD. Before analysis, all samples were attached to stubs with double-sided adhesive tape. Subsequently, all samples were made electrically conductive by coating them with platinum at a rate of 6 nm/min in a vacuum (7 × 10^−3^ mbar) using a Sputter Coater (K575X; EmiTech, Madrid, Spain). The coated samples were placed in a scanning electron microscope to observe their surface morphological characteristics [40].

#### 2.6.2. Thermal Properties

To examine the thermal properties of OLA-SD, thermal analysis was performed using differential scanning calorimetry with a DSC Q200 (TA Instruments, New Castle, DE, USA).

Weighed samples, approximately 3–5 mg each, were placed in standard aluminum pans, and nitrogen was used as the purge gas. All samples were scanned at a temperature ramp rate of 10 °C/min, and heat flow was measured from 20 °C to 260 °C [41].

#### 2.6.3. Crystallinity State

The crystallinity of the manufactured OLA-SD was inspected using a powder X-ray diffractometer (D/MAX-2500; Rigaku, Tokyo, Japan). During the analysis, PXRD patterns were recorded using Cu Kα radiation (λ = 1.54178 Å) at a power setting of 40 kV and 100 mA. An angular increment of 0.02° per second was selected for scanning the 2θ angle range from 2° to 60° [42].

#### 2.6.4. Fourier Transform Infrared Spectroscopy Analysis

To examine the infrared spectrum of OLA-SD, Fourier Transform Infrared (FT-IR) spectroscopy analysis was performed using a Spectrum Two^TM^ (PerkinElmer, Waltham, MA, USA). The KBr pellet was prepared by mixing approximately 1 mg of the sample with 200 mg of KBr and using a pellet press. The samples were scanned across a spectral range from 4000 to 400 cm^−1^ [43].

### 2.7. In Vitro OLA-SD Release

To assess the in vitro release of OLA from the solid dispersions, a dissolution test was performed under conditions similar to those described in Section 2.3. Briefly, the dissolution media were adjusted to pH 1.2 and pH 6.8 (300 mL each) [44,45,46], with the temperature maintained at 37 ± 0.5 °C, and the test was conducted at a paddle speed of 50 rpm. Each sample of the OLA-SD, equivalent to 150 mg of OLA, was immersed in the dissolution media. Additionally, to examine the effect of polymers in the solid dispersion, a parallel dissolution test was conducted with 150 mg each of crystalline OLA and amorphous OLA. As mentioned, the solution was sampled at predetermined times (1, 6, 12, 18, 24, 48, and 72 h) and the concentration of OLA was assessed using an HPLC system. Additionally, after the kinetic dissolution tests in the pH 1.2 solution, the solution was centrifuged, the supernatant was removed, and the remaining OLA-SD residue was dried overnight for 12 h. The crystallinity state of the dried OLA-SD was examined using an X-ray diffractometer. The detailed method is described in Section 2.6.3.

### 2.8. Morphological Changes in OLA-SD Dispersed in Aqueous Solution

The morphological characteristics of two selected OLA-SDs, Kollidon VA64 (F1) and HPMC P645 (F3), were examined over time using scanning electron microscopy after immersion in D.W. at 37 ± 0.5 °C [47]. Approximately 20 μL of solution samples were withdrawn every 24 h, and each sample was placed on a glass slide. The glass slides were dried overnight in a vacuum oven set to 25 °C for approximately 12 h. The dried glass slides were then coated with platinum for imaging. Detailed SEM methods are described in Section 2.6.1.

### 2.9. In Vivo Pharmacokinetic Study

Male Sprague Dawley (SD) rats (age: 9–10 weeks, weight: 250 ± 20 g) for the in vivo pharmacokinetic study of OLA-SD were purchased from Koatech Co. (Pyeongtaek, Republic of Korea). The rats were acclimatized to standard laboratory conditions (average temperature of 25 ± 2 °C and a 12/12 h light/dark cycle) for one week before the experiment, with unrestricted access to food and drinking water ad libitum. The Institutional Animal Care and Use Committee (IACUC) of Gyeongsang National University (Approval No. GNU-240228-R0049) approved the animal study protocol in accordance with NIH policies and the Animal Welfare Act. The SD rats were randomly divided into four groups, each consisting of six rats. Each rat was anesthetized, and the right femoral artery was cannulated using a polyethylene tube. Crystalline OLA, amorphous OLA, Kollidon VA64-based OLA-SD (F1), and HPMC P645-based OLA-SD (F3) were dispersed in 1 mL of 0.5% carboxymethyl cellulose and administered orally to the rats at a dose of 20 mg/kg. Blood (350 μL) was collected through the cannulated tube at 0.25, 0.5, 1, 2, 3, 6, 9, 12, and 24 h and immediately centrifuged at 13,500 rpm for 15 min at 4 °C. Subsequently, plasma was separated from the supernatant and stored at −20 °C until quantitative analysis. To analyze the plasma samples, 50 μL of internal standard solution (canagliflozin 20 μg/mL in acetonitrile) and 300 μL of acetonitrile were added to 150 μL of plasma. The samples were then mixed in a vortex blender for 3 min for deproteinization and drug extraction. Subsequently, the samples were centrifuged at 13,500 rpm for 15 min at 4 °C. The supernatant was filtered using a 0.2 μm filter and transferred to analytical vials for HPLC analysis. The concentration of OLA in plasma was analyzed using the second HPLC conditions mentioned in Section 2.2. Pharmacokinetic parameters calculated using non-compartmental analysis included the area under the plasma concentration–time curve from 0 to 24 h (AUC_0–24_), the maximum plasma concentration (C_max_), the time to reach maximum plasma concentration (T_max_), the elimination rate constant (K_el_), the half-life (T_1/2_), and the mean residence time (MRT).

The formula for calculating AUC is as follows:$$\mathrm{AUC}=\sum_{i=1}^{n-1}\frac{\left(C_{i}+C_{i+1}\right)}{2}\times (t_{i+1}+t_{i})$$
where C_i_ and C_i+1_ are the concentration of the drug at consecutive time points, t_i_ and t_i+1_ are the corresponding time points, and n is the number of measured data points.

The formula for calculating C_max_ is as follows:$$C_{\max}=\max(C\left(t\right))$$
where C_t_ is the concentration of the drug at time t.

The formula for calculating T_max_ is
$$T_{\max}=t\;\mathrm{at}\;\mathrm{which}\;C\left(t\right)\;\mathrm{is}\;\mathrm{maximal}$$

The formula for calculating K_el_ is
$$K_{\mathrm{el}}=\frac{\ln\left(C_{2}\right)-\ln(C_{1})}{t_{2}-t_{1}}$$
where C_1_ and C_2_ are the concentration of the drug at two consecutive time points during the elimination phase, and t_1_ and t_2_ are the corresponding time points.

The formula for calculating T_1/2_ is
$$T_{1/2}=\frac{\ln\left(2\right)}{K_{\mathrm{el}}}$$
where K_el_ is the elimination rate constant.

The formula for calculating MRT is
$$\mathrm{MRT}=\frac{\mathrm{AUMC}}{\mathrm{AUC}}$$
where AUMC is the area under the moment curve and AUC is the area under the curve.

### 2.10. Statistical Analyses

All data were expressed as mean ± SD (standard deviation), and regression analysis of variance (ANOVA) was performed using Minitab ver. 19 software (Minitab Inc., State College, PA, USA) to calculate the mean and standard deviation of each test group and to verify the statistically significant differences between each test group at a significance level of 5% (*p* < 0.05).

## 3. Results

### 3.1. Solubility of OLA

The solubility of a drug is a key parameter that affects drug absorption and bioavailability [48]. The saturation solubility and kinetic solubility of crystalline OLA and amorphous OLA were assessed in solutions of pH 1.2, pH 4.0, and pH 6.8, which represent physiological pH conditions (Figure 2 and Figure 3) [49]. The saturation solubility of both crystalline OLA and amorphous OLA was consistently very low, at around 100 µg/mL, regardless of the pH (Figure 2).

However, the results of kinetic solubility tests showed that amorphous OLA had higher solubility than crystalline OLA and both amorphous OLA and crystalline OLA showed pH-independent solubility (Figure 3). These results suggested that the enthalpy, entropy, and free energy of amorphous drugs are higher compared to their crystalline counterparts, contributing to improved solubility [50]. In contrast, despite the high-energy characteristics of amorphous OLA, which contribute to its initially higher solubility, the reason the saturation solubility of amorphous OLA remains similarly low to that of crystalline OLA is that it tends to transform into a crystalline state over time, thereby losing its solubility advantage [51]. If the amorphous drug recrystallizes and does not dissolve in the gastrointestinal tract, its bioavailability may decrease [52]. Therefore, the solubility results for OLA underline the importance of selecting polymers that can maintain a stable amorphous form and reduce the rate of recrystallization.

### 3.2. Selection of Polymers for the Solid Dispersion

To select an appropriate polymer, the saturation solubility of crystalline OLA in 1% (*w*/*v*) polymer solutions was tested across a total of 22 types of polymers widely used in solid dispersions (Figure 4). As a result, the saturation solubility of OLA in all polymer solutions, except for HP-β-CD, did not significantly improve, with no notable differences observed among the types of polymers. Among them, the saturation solubility of OLA in HP-β-CD was the highest, reaching approximately 179.9 ± 6.7 μg/mL. This increase is attributed to the unique molecular structure of HP-β-CD, which features a hydrophobic internal cavity and a hydrophilic external surface capable of encapsulating poorly soluble drugs, thereby enhancing their solubility [53]. For this reason, a solid dispersion system incorporating HP-β-CD is widely used to enhance the solubility of poorly water-soluble drugs [54]. However, in most other polymer solutions, the saturation solubility of OLA did not reach high levels, remaining below 150 μg/mL.

Based on these results, and considering the low solubility of crystalline OLA, the kinetic solubility of both crystalline and amorphous forms was compared (Figure 5). To investigate kinetic solubility, selected polymer solutions included povidone derivatives (Kollidon VA64, PVP K-30), cellulose derivatives (HPMC P645, HPC L-type), HP-β-CD, and PEG6000, which are widely used in solid dispersion systems [55]. The kinetic solubility results for crystalline OLA showed that although solubility did not reach very high levels in any of the six types of polymer solutions, it gradually increased over time (Figure 5A). Similar to the saturation solubility results shown in Figure 3, the highest solubility was observed in HP-β-CD. However, the kinetic solubility of amorphous OLA was higher than that of crystalline OLA in all polymers, displaying a different pattern of results (Figure 5B). As mentioned above, the overall solubility was improved because the amorphous form exhibits higher solubility compared to the crystalline form. Specifically, in Kollidon VA64, the solubility of OLA was 592.6 ± 72.9 μg/mL at 1 h, which was higher than in other polymers, but it gradually decreased to 343.2 ± 6.5 μg/mL at 72 h. These results are attributed to amorphous OLA exceeding the solubility of the crystalline form, thereby increasing the drug concentration through the occurrence of a supersaturated solution, but then losing the solubility advantage as the drug recrystallizes, which is a major disadvantage of supersaturated solutions [56]. In the case of PVP K-30 and PEG6000, similar to Kollidon VA64, the solubility of OLA was high initially but gradually decreased over time. Conversely, in HPMC P645, the solubility of OLA was 247.2 ± 43.6 μg/mL at 1 h and increased to 353.4 ± 44.2 μg/mL at 72 h, showing an increase in solubility over time, unlike the pattern observed with Kollidon VA64. HPC L-type also exhibited a similar pattern to HPMC P645, with solubility gradually increasing over time. These results could be supported by the finding that cellulose derivatives, due to their characteristics such as physiological conditions, strong drug–polymer interactions, and high glass transition temperatures (Tg), are primarily used for the stabilization of amorphous drugs [57]. The solubility of OLA in HP-β-CD was 336.7 ± 29.7 μg/mL at 1 h, showing the second highest solubility after Kollidon VA64, and did not significantly decrease over time, recording 397.9 ± 10.9 μg/mL at 72 h. To improve the solubility and stability of OLA-SD, it is important to have excellent miscibility and compatibility between the OLA and the polymers [58]. Therefore, the selection of appropriate polymers is related to enhancing the bioavailability of OLA [59]. Considering the overall solubility results of OLA, the povidone derivatives, including Kollidon VA64 and PVP K-30, effectively reached a supersaturation state initially and demonstrated high solubility. Additionally, HPMC P645 and HP-β-CD stably maintained the amorphous state without reducing the solubility of OLA. Based on these results, OLA-SD was prepared using Kollidon VA64 and PVP K-30, which effectively increased the solubility of amorphous OLA, and HPMC P645 and HP-β-CD, which stably maintained the amorphous state. The characteristics of each formulation were evaluated, and the oral absorptions of the selected formulations were compared through in vitro evaluation of OLA-SD.

### 3.3. Physicochemical Characterization of OLA-SD

The physicochemical characteristics of the prepared OLA-SDs were evaluated. SEM images of crystalline OLA and OLA-SDs are shown in Figure 6. The crystalline OLA exhibited an irregularly shaped polygonal crystal structure, with particle sizes of approximately 10 μm. In contrast, OLA-SD showed significant changes in particle shape and surface morphology. In OLA-SD, no crystalline structure was observed, and the particles exhibited a smooth and spherical shape. This indicates that the crystalline OLA was transformed into spherical particles through spray drying technology.

Figure 7A shows the thermal behaviors of crystalline OLA, amorphous OLA, and OLA-SDs. The crystalline OLA exhibited a sharp endothermic peak at 213 °C, consistent with previously reported results [60], indicating crystalline characteristics. In contrast, the intrinsic peaks disappeared in amorphous OLA and OLA-SD, indicating that OLA-SD is amorphous and lacks a regular crystalline structure, which leads to the absence of endothermic peaks [61]. The powder X-ray diffraction (PXRD) patterns are shown in Figure 7B. High-intensity peaks were exhibited at different diffraction angles in the crystalline OLA. However, these high-intensity peaks observed in the crystalline OLA disappeared in amorphous OLA and OLA-SD, indicating that the drug is in an amorphous state [62]. Consequently, it was observed that the OLA-SD formulation had transformed from a crystalline state to a high-energy amorphous state through spray drying technology. In this study, FT-IR was used to investigate intermolecular interactions by identifying molecular stretching vibrations or peak broadening. The FT-IR spectra are shown in Figure 7C. In the case of crystalline OLA, a strong absorption band was observed at 3400 cm^−1^ (N-H, amide), 3000–3165 cm⁻^1^ (N-H, amine or amide), 1611–1655 cm^−1^ (C=O, carbonyl), 1400–1600 cm^−1^ (C=C, aromatic), and 750–812 cm^−1^ (C-H, aromatic) [63,64]. Similarly, OLA-SD exhibited the characteristic stretching bands observed in crystalline OLA, indicating that there were no intermolecular interactions in the solid dispersions. In conclusion, based on the results of SEM, DSC, PXRD, and FT-IR, it was observed that all OLA-SDs exhibited smooth, spherical particles in an amorphous state, with no molecular alterations in OLA within the solid dispersions.

### 3.4. Dissolution Profile of Prepared OLA-SDs

In vitro dissolution tests for OLA-SDs were performed in pH 1.2 and pH 6.8 solutions, simulating gastrointestinal pH conditions (Figure 8). To compare the dissolution behaviors, tests were also conducted with crystalline OLA and the pure amorphous form of OLA without a polymer. The dissolution rate of OLA-SD was found to be pH-independent due to the fundamental solubility characteristics of OLA and the use of hydrophilic polymers that dissolve well in aqueous solutions [65,66,67,68]. All OLA-SD formulations demonstrated higher dissolution rates compared to crystalline OLA, regardless of pH. Specifically, the dissolution rate of amorphous OLA reached 43.4 ± 3.3% and 44.6 ± 3.1% at pH 1.2 and pH 6.8 at 1 h, respectively, decreasing to 28.2 ± 3.9% and 27.0 ± 3.0% at 72 h. This decrease may be attributed to the inherent tendency of amorphous drugs to crystallize from a supersaturated solution [69]. Pure amorphous OLA without polymers showed less effective maintenance of a supersaturated state compared to OLA-SDs, resulting in decreased solubility [70]. However, amorphous OLA-SDs containing polymers exhibited higher dissolution rates than amorphous OLA without polymers. These results suggest that a molecularly dispersed blend of drug and polymer in an amorphous formulation enhances solubility compared to a crystalline drug and reduces the propensity for the drug to crystallize from a supersaturated state [71]. Notably, F1, based on the povidone derivative Kollidon VA64, achieved the highest dissolution rates at pH 1.2 and pH 6.8 with 97.7 ± 1.6% and 93.1 ± 1.8% at 6 h, respectively. However, a rapid decrease in dissolution rates was observed after 6 h, falling to 40.3 ± 0.4% and 41.5 ± 0.6% at 72 h due to recrystallization. PVP K-30-based F2 also exhibited higher dissolution rates between 1 and 6 h compared to other OLA-SDs but showed a similar trend of decreasing rates over time. F3, based on the cellulose derivative HPMC P645, did not exhibit high initial dissolution rates compared to the povidone-based F1 and F2. However, unlike F1 and F2, the dissolution rate of F3 did not decrease over time but gradually increased, achieving 83.7 ± 4.1% at pH 1.2 and 82.6 ± 4.6% at pH 6.8 at 72 h. These results were consistent with previous reports, which indicated that HPMC was more effective than Kollidon VA64 in inhibiting drug recrystallization in amorphous solid dispersions [72]. Interactions between the drug and polymer can influence the physical stability of solid dispersions, with the formation of hydrogen bonds between the molecularly dispersed drug and polymer being a critical factor for stability [73]. The carbonyl group (C=O) and amide group (N-H) of OLA function as hydrogen bond acceptors [74,75]. Unlike povidone derivatives, HPMC P645 contains multiple hydroxyl groups, which can enhance the stability of solid dispersions through hydrogen bonding with OLA [76]. Moreover, polymers with a high glass transition temperature offer an approach to maintaining a stable amorphous form [77]. HPMC, typically exhibiting a higher glass transition temperature of around 160–210 °C, may be more effective than Kollidon VA64, which has a glass transition temperature of approximately 106 °C [78,79]. The dissolution rate of the HP-β-CD based solid dispersion F4 was approximately 50% from 1 h to 72 h, regardless of pH, which is lower compared to F1–F3, and it was observed to gradually decrease over time. Despite the high solubility of OLA in a 1% (*w*/*v*) HP-β-CD solution, the lower solubility rate of F4 is assumed to be due to the insufficient complexation of the drug with HP-β-CD at low concentrations. Based on the dissolution results, OLA-SDs showed a higher dissolution rate compared to both crystalline OLA and amorphous OLA. Particularly, the F3 formulation based on HPMC P645 maintained a high dissolution rate and did not display recrystallization over time. These results indicate that by improving the low solubility of OLA, enhanced oral absorption can be achieved.

After the kinetic dissolution tests in the pH 1.2 solution, the solution was centrifuged, the supernatant was removed, and the remaining OLA-SD residue was dried overnight for 12 h. The dried residue was then analyzed for its crystallinity using an X-ray diffractometer (Figure 9). Analysis of the PXRD patterns revealed that the intensity values increased at specific theta angles characteristic of OLA for F1. In contrast, for F3, the intensity values did not increase at specific theta angles of OLA, indicating that OLA did not recrystallize and precipitate in the aqueous solution.

### 3.5. Comparison of the Characteristics of OLA-SDs Dispersed in Aqueous Solution

The morphological differences between the two OLA-SDs, F1 (based on Kollidon VA64) and F3 (based on HPMC P645), which exhibit contrasting dissolution patterns, were observed through SEM images after dispersing them in an aqueous solution (Figure 9). In the case of F1, the sample almost completely dissolved within 5 min, appearing clear upon visual inspection. This observation was corroborated by SEM images, where no distinct particles were visible at the 5 min mark (Figure 10A). However, by the 1 h mark, small needle-shaped particles began to appear, and by 24 h, more distinct and numerous needle-shaped particles were observed. Over time, the initially clear solution gradually became turbid as the sample precipitated, a pattern confirmed through SEM images. In contrast, F3 displayed initially undissolved particles when dispersed in the aqueous solution. SEM images taken at the 5 min mark supported the presence of these undissolved particles in a spherical form (Figure 10B). These spherical particles remained clustered together over time, unlike the needle-shaped particles observed in F1. This result indicated that the needle-shaped particles observed in the SEM images had precipitated due to the recrystallization of OLA. These overall results from SEM and PXRD showed a correlation with the dissolution patterns. The dissolution data reveal that F1 undergoes rapid dissolution initially, evidenced by a substantially elevated dissolution rate. However, this rate declines over time in the aqueous solution due to the recrystallization of OLA. In contrast, F3, which does not completely dissolve, exhibits a lower initial dissolution rate than F1. Despite this, the dissolution rate of F3 remains consistent and stable over time, which is attributable to the absence of OLA recrystallization.

### 3.6. In Vivo Pharmacokinetic Study

The in vivo pharmacokinetic behavior of OLA-SD (F1, F3), amorphous OLA, and crystalline OLA was investigated in rats. The mean plasma concentration–time profiles of OLA and the corresponding pharmacokinetic parameters are depicted in Figure 11 and Table 2, respectively. The AUC_0–24_ and C_max_ for crystalline OLA were 370.11 ± 63.75 ng/mL and 187.31 ± 82.55 ng·h/mL, respectively. In comparison, the AUC_0–24_ and C_max_ for amorphous OLA were 471.40 ± 150.17 ng/mL and 415.24 ± 148.20 ng·h/mL, respectively, representing increases of 1.27-fold and 2.22-fold over the crystalline OLA. AUC_0–24_ and C_max_ for the F1 formulation were 995.38 ± 194.76 ng/mL and 1459.05 ± 548.86 ng·h/mL, respectively, which were 2.69-fold and 7.79-fold higher than those of the crystalline OLA. Similarly, the AUC_0–24_ and C_max_ for the F3 formulation were 1551.47 ± 484.09 ng/mL and 2001.25 ± 734.43 ng·h/mL, respectively, achieving 4.19-fold and 10.68-fold increases. Solid dispersions F1 and F3 exhibited higher AUC_0–24_ and C_max_ levels compared to amorphous OLA without polymers. The F3 formulation especially demonstrated higher plasma concentrations than the F1 formulation, with AUC_0–24_ and C_max_ values being 1.56-fold and 1.37-fold higher, respectively. As observed in the in vitro study, the F1 formulation initially showed a higher release rate compared to the F3 formulation but exhibited rapid recrystallization, which was also confirmed by changes in appearance observed through SEM and PXRD. In contrast to the F1 formulation, the F3 formulation showed a high release rate and maintained its amorphous state stably over time without recrystallization, as confirmed by SEM and PXRD. Based on these results, considering that the F3 formulation showed higher AUC_0–24_ and C_max_ levels compared to the F1 formulation, it is important to select polymers that prevent recrystallization and maintain a stable amorphous state. This study demonstrated that HPMC-based OLA-SDs can effectively enhance the oral absorption of OLA.

## 4. Conclusions

In this study, we investigated the formulation of stable solid dispersions to enhance the bioavailability of OLA, a therapeutic agent for ovarian cancer characterized as a BCS class IV drug with low solubility and low permeability. By utilizing polymers, we aimed to improve the solubility of OLA and, thereby, its bioavailability. Evaluations of solubility and kinetic solubility revealed that while amorphous OLA showed an improvement in kinetic solubility compared to its crystalline counterpart, the saturation solubility values were similar. These findings indicated that the solubility patterns of OLA in polymer solutions vary according to the type of polymer used. Among the selected polymers, the povidone derivative Kollidon VA64 demonstrated excellent initial release rate improvement; however, a reduction in solubility over time was observed. In contrast, the cellulose derivative HPMC P645 exhibited a sustained improvement in solubility. Through screening various polymers, Kollidon VA64, PVP K-30, HPMC P645, and HP-β-CD were selected to fabricate solid dispersions containing OLA, and their physicochemical properties were evaluated. Among the manufactured OLA-SDs, HPMC P645 was chosen as the polymer that could maintain a stable amorphous state without reducing the solubility of OLA. OLA-SD containing HPMC P645 continuously maintained the amorphous state in a supersaturated solution, exhibiting high solubility without a decrease in dissolution rate. No significant morphological changes or recrystallization of the particles were observed in the aqueous solution. Furthermore, the in vivo study demonstrated that the HPMC P645-based OLA-SD formulation exhibited higher bioavailability compared to crystalline OLA, amorphous OLA, and Kollidon VA64-based OLA-SDs. In conclusion, this formulation was developed as a viable option to improve the solubility and bioavailability of poorly soluble OLA. The spray drying technology used in this study to prepare HPMC-based OLA-SD is compatible with current pharmaceutical production techniques, ensuring the feasibility of large-scale production. Based on the improved oral absorption of this formulation, it has the potential to be industrially applied, improving patient outcomes with more effective treatments for ovarian cancer, breast cancer, and other diseases.

## Data Availability

Data are available on request due to restrictions, e.g., privacy or ethical restrictions.

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
