# Peer review of "Investigation of Stabilized Amorphous Solid Dispersions to Improve Oral Olaparib Absorption"

_pharmaceutics, 2024, doi:10.3390/pharmaceutics16070958_

Round 1

Reviewer 1 Report

Comments and Suggestions for Authors

The current study aimed to formulate olaparib-containing solid dispersions in order to enhance the oral bioavailability of this drug characterized by low solubility and low permeability. Among 22 types of polymers tested, 4 of them were selected for the formulations of solid dispersions, and 2 of them were selected for in vivo animal pharmacokinetic study, where hypromellose (HPMC P645) proved to be the most beneficial since it stabilizes the amorphous structure of the drug. Generally, this comprehensive original study is very well designed, the methods are contemporary and appropriate for this type of a study, and the obtained results are new and significant, which brings novelty to the research field. There are a few concerns and suggestions that should be addressed.

Why did you choose the ratio 1:2 (150 mg OLA + 300 mg polymer) for all solid dispersions? It should be stated in the text as well.

Two similar HPLC methods were used for measuring drug concentrations. However, none of them were referred to. If they are new methods, the validation parameters need to be specified in the text.

Olaparib is an international nonproprietary name (INN) of a drug and thus should be written with the first lowercase letter throughout the text. Similarly, hypromellose should be written with the first lowercase letter (line 15), as well as ethanol (line 75).

The types of carbomers should be specified in the ‘Materials’ section since there are two synthalens in the Results.

It is not sufficient to state only pH value of aqueous solutions (pH 1.2, 4.0, 6.8), it should be specified if it is a buffer, if so – which one and at which concentration.

Line 152: Why 300 mL?

Figure 2: Why is the concentration range at y-axis up to 10,000 µg/mL?

Figure 3: Time at x-axis is not correct. Besides, it is important to conclude that solubility is not pH-dependent. It was stated that the dissolution rate of OLA-SD was pH-independent; however, this should be stated also for the solubility.

Comments on the Quality of English Language

No major concerns were detected.

Reviewer 2 Report

Comments and Suggestions for Authors

Kyeong Soo Kim et al. reported an interesting work upon ASD for olaparib administration. The topic was of a certain significance, and fell within the scope of Pharmaceutics. However, there were some issues hampering the acceptance of this submission. Please consider to perform a Major Revision before a second consideration. Detailed comments:

1.      The Introduction should not be written in a single paragraph. The current version read too long. Please consider to split to 2~3 paragraphs, and draw an illustration about the ASD design at the end of it.

2.      Please notice that Pharmaceutics is a flagship journal its field, which emphasizes the use of statistical analysis. The relevant information should be demonstrated in the Method Section.

3.      According to Figure 2, the saturation solubility of ASD was even a bit lower than the drug powder. What was the reason? Typically, ASD had a significantly higher solubility than drug powder.

4.      For Figure 3, could the authors transform into accumulated drug release ratio?

5.      The characteristic peaks should be marked for olaparib, in Figure 7.

6.      Could the authors also calculate MRT in Table 2?

7.      Please make some comments about the industrialization aspects before the Conclusion Section.

Reviewer 3 Report

Comments and Suggestions for Authors

The current research manuscript focusing on the investigation of Olaparib solid dispersion developed by spray drying approach for improved bioavailability is interesting and matches the scope of the journal. However, the methodologies were incomplete with some missing information. Authors are requested to address the below comments for improving the quality of the manuscript:

1.     Please provide information for drug solubility in water and whether its acidic or basic, also provide information sushc as pKa, and logP.

2.     Is olaparib commercially approved for oral administration? Please make a note of commercial product and the dose.

3.    Section 2.2: What type of filter was used? Hydrophilic or hydrophobic ? Line 98: what are the predetermined time points? How was drug added to the dissolution vessel? was any capsules used? Please describe HPLC methodology as a separate section before section 2.2. Also cite any reference if followed, if its an in-house developed method, please mention LOQ and LOD. Please provide information for calibration curve.Why 150 mg dose was chosen for kinetic solubility? Does the media volume (900 mL) is providing sink condition or non-sink condition? Was kinetic solubility performed in all the pH medias?

4.    Section 2.3: were samples collected at different time points only for kinetic solubility?

5.    Section 2.4: Following spray drying, was the product dried further? How did authors ensure complete drying?

6.    Table 1: write down the units

7.    Section 2.6: How was pH adjusted? was dissolution also studies in water? How was sample added to the dissolution media? what are the predetermined time points?

8.    Section 2.5.2: Approximately how much quanitity of sample was sealed in the aluminum pans and what type of pans was used? How was data interpreted?

9.    Section 2.7: what was the temperature of the vacuum oven?

10.Section 2.8.3: The title doesn't suite the methodology described here. Please remove the subsections under 2.8 and make it as a single methodology. Also provide the formulas for pharmacokinetic parameters.

Comments on the Quality of English Language

Moderate English editing is needed.

Reviewer 4 Report

Comments and Suggestions for Authors

This study investigated the in vitro solubility characteristics of the Olaparib (OLA) drug powder and amorphous OLA under different pH conditions. Thus, emphasis was placed on the challenges faced in maintaining the amorphous form under a wide range of pH conditions. These in vivo studies in rats revealed that the solid dispersion formulations with the identified polymers, Kollidon VA64 and HPMC P645, drastically improved the solubility and effectively prevented the recrystallization of OLA, in turn enhancing oral bioavailability in rats. Overall, the study highlighted how polymer selection and formulation design are essential to augment the bioavailability of poorly water-soluble drugs, such as OLA. A few remarks regarding the authors' considerations are as follows:

Introduction:

1. In several instances the writing needs some attention. Some of these sentences are long and complicated and require shortening to be comprehensive. Example: "Ovarian cancer is associated with a low survival rate; only 50% of patients diagnosed with ovarian cancer in the United States survived for five years after diagnosis (2012–2018)."

2. Connect the link between the problem (low solubility and permeability of OLA) and the chosen solution (solid dispersion). Sample: Explain why solid dispersion is preferred over the other techniques.

3. There is no work dealing with OLA-loaded solid dispersions, which is a significant gap in the literature.

Methods:

4. Rationale for Selection of the Polymers: This section can be enhanced by explaining the selection of polymers to be used within the solid dispersion system. It would provide a much stronger scientific basis if the rationale for the selection over other polymers was explained based on preliminary data or the literature.

5. The methods used to prepare amorphous OLA are cited only in this section but without a description. Mentioning the details here would be more comprehensive for the reader to understand the methodology.

6. This appears to be a general section that lacks justification for the chosen methods. For instance, why is spray drying the process of solid dispersion preparation? This would help in discussing the advantages of spraying in this context.

Results and discussion

7. While this section provides specific experimental results, the implications for OLA formulation development need to be made more explicit.

8. More discussion can be made on the selection of polymers for the obtained solid dispersion, providing some rationale for the choice of specific polymers and their potential impact on stability and bioavailability. The explanation for the in vivo pharmacokinetic studies can be further expanded by discussing how the formulation characteristics observed in the in vitro study translated into enhanced oral absorption in vivo.

Reviewer 5 Report

Comments and Suggestions for Authors

Dear authors,

I have reviewed your manuscript titled “Investigation of Stabilized Amorphous Solid Dispersions for  Improved Oral Absorption of Olaparib” submitted to Pharmaceutics journal. I would like to commend you on the quality and thoroughness of your research. Your study offers significant insights and makes a valuable contribution to the field. While the manuscript is overall well-prepared, I have identified a minor revisions that would improve its clarity and overall impact:

1-I suggest the authors to change the title "Investigation of Stabilized Amorphous Solid Dispersions for Improved Oral Absorption of Olaparib" in "Investigation of Stabilized Amorphous Solid Dispersions to improve Olaparib Oral Absorption".

2- The first part of the introduction, from lines 25 to 27, discussing ovarian cancer, is not necessary. Since the focus of the article is on increasing the solubility and bioavailability of Olaparib, and the rat used in the in vivo experiment did not have ovarian cancer, it would be better to delete this part.

3- I suggest the authors to include studies on medication with similar characteristic to Olaparib, low in solubility and bioavalibility, which have been developed using solid dispersion method.

4- In line 47, mention OLA-SD after OLA- loaded solid dispersion.

5- In line 28, delete the Olaparib.

6- In lines 54 and 55, delete the scanning electron microscopy, differential scanning calorimetry, powder X-ray diffraction, Fourier transform infrared spectroscopy.

7- In all figures and graphs, replace "OLA drug powder" with "crystalline OLA." Standardize the captions and labels by consistently using "crystalline OLA" instead of "OLA drug powder.

8- In line 231, it is written that 22 types of polymers, while in the figure 4 there are 23 types.

9- In line 245, it is better to say ’to investigate kinetic solubility, selected polymer solutions included povidone derivatives (Kollidon VA64, PVP K-30), cellulose derivatives (HPMC P645, HPC L-type), HP-β-CD, and PEG6000, which are widely used in solid dispersion systems.’ For more clarification.

10- In line 282, delete Scanning electron microscope.

11- In the results section, titled "Physicochemical Characterization of OLA-SD," it would be helpful to specify what F1-F4 are. Although this is explained in the materials and methods section, mentioning it again here would provide additional clarity.

12- For the differential scanning calorimetry thermograms, powder X-ray diffractograms, and Fourier transform infrared spectrometer analyses, it would be beneficial to include the amorphous form of Olaparib.

13- In the caption of figure 7, replace PXRD with powder x-ray diffractogram.

Reviewer 6 Report

Comments and Suggestions for Authors

This manuscript "Investigation of Stabilized Amorphous Solid Dispersions for

Improved Oral Absorption of Olaparib" is overall good and covers important area. The authors investigated the formulation of stable solid dispersions to enhance the bioavailability of Olaparib (OLA), a therapeutic agent for ovarian cancer characterized as a BCS Class IV drug with low solubility and low permeability. I suggest the following corrections:

1- The title is too long and better to be shortened.

2- The most important numerical findings needed to be included in the abstract part.

3- Please provide more details regarding the characterization methods.

4- Please make a graphical abstract if possible.

5- Have you evaluated the loading efficiency of the prepared formulations?

6- The authors have mentioned that the dissolution rate of amorphous OLA reached 43.4 ± 3.3% and 44.6 ± 3.1% at pH 1.2 and pH 6.8 respectively at 1 hour. How did you know that it is 43.4% or 44.6%? Are you sure that 150 mg of Ola is successfully loaded?

7- Regarding the dissolution profiles, I suggest the authors to fit these data with the Korsmeyer-Peppas model and/or other kinetic models for further investigations. You can use this article as a reference for this modeling: https://doi.org/10.3390/gels9120929

Comments on the Quality of English Language

Minor.

Round 2

Reviewer 2 Report

Comments and Suggestions for Authors

Thanks for your revision. I notice that you did not respond to Question 6 in your response. Maybe there are some difficulties, but I suggested you to try to express you own opinions on the quesions even if you cannot fix it, in your future submissions.

Reviewer 3 Report

Comments and Suggestions for Authors

All the comments are well addressed with proper justification. The revised version of the manuscript can be accepted for publication.

Reviewer 4 Report

Comments and Suggestions for Authors

No comments!

Reviewer 6 Report

Comments and Suggestions for Authors

The authors have addressed all comments successfully.

Comments on the Quality of English Language

Minor editing